# Say NO to ROS: Their Roles in Embryonic Heart Development and Pathogenesis of Congenital Heart Defects in Maternal Diabetes

**DOI:** 10.3390/antiox8100436

**Published:** 2019-10-01

**Authors:** Anish Engineer, Tana Saiyin, Elizabeth R. Greco, Qingping Feng

**Affiliations:** Department of Physiology and Pharmacology, Schulich School of Medicine and Dentistry, Western University, London, Ontario, ON N6A 5C1, Canada; aengine@uwo.ca (A.E.); tsaiyin@uwo.ca (T.S.); egreco5@uwo.ca (E.R.G.)

**Keywords:** congenital heart defects, pregestational diabetes, nitric oxide, reactive oxygen species, eNOS, heart development, tetrahydrobiopterin, oxidative stress, antioxidant, exercise

## Abstract

Congenital heart defects (CHDs) are the most prevalent and serious birth defect, occurring in 1% of all live births. Pregestational maternal diabetes is a known risk factor for the development of CHDs, elevating the risk in the child by more than four-fold. As the prevalence of diabetes rapidly rises among women of childbearing age, there is a need to investigate the mechanisms and potential preventative strategies for these defects. In experimental animal models of pregestational diabetes induced-CHDs, upwards of 50% of offspring display congenital malformations of the heart, including septal, valvular, and outflow tract defects. Specifically, the imbalance of nitric oxide (NO) and reactive oxygen species (ROS) signaling is a major driver of the development of CHDs in offspring of mice with pregestational diabetes. NO from endothelial nitric oxide synthase (eNOS) is crucial to cardiogenesis, regulating various cellular and molecular processes. In fact, deficiency in eNOS results in CHDs and coronary artery malformation. Embryonic hearts from diabetic dams exhibit eNOS uncoupling and oxidative stress. Maternal treatment with sapropterin, a cofactor of eNOS, and antioxidants such as N-acetylcysteine, vitamin E, and glutathione as well as maternal exercise have been shown to improve eNOS function, reduce oxidative stress, and lower the incidence CHDs in the offspring of mice with pregestational diabetes. This review summarizes recent data on pregestational diabetes-induced CHDs, and offers insights into the important roles of NO and ROS in embryonic heart development and pathogenesis of CHDs in maternal diabetes.

## 1. Introduction

Congenital heart defects (CHDs) are the most common structural birth defect, occurring in 1–5% of live births [1,2,3]. Congenital heart disease is the leading cause of pediatric deaths in developed nations [4]. Pregestational diabetes is an important risk factor of CHDs [5]. While good glycemic control in diabetic mothers lowers the risk, the incidence of CHDs in their children is still higher than in the general population [6]. Congenital malformations in the offspring of women with pregestational diabetes include defects of the limb, neural tube, and musculoskeletal systems. However, the predominant malformations are CHDs, which represent about 40% of the total malformations [7,8]. Currently, 40% of women with diabetes are of reproductive age [9]. The rapid increase of young adults with diabetes and prediabetes is alarming, and warrants further research to broaden our understanding of the pathogenesis of congenital cardiovascular malformations and their prevention.

To understand pathogenesis of CHDs in maternal diabetes, various experimental approaches have been employed. In animal models, diabetes can be induced via many techniques, including diet, drugs, surgery, or genetic alterations. A common, established method for the generation of a type 1 diabetes (T1D) model in rodents is streptozotocin (STZ) administration [10,11]. This chemical agent is a glucose analogue, which is taken up by pancreatic β-cells via the glucose transporter 2 (GLUT2). STZ within the cell causes DNA damage, oxidative stress, and ultimately results in β-cell death [12]. This depletion of pancreatic β-cells and thereby insulin supply recapitulates T1D in humans. Animal studies analyzing heart development in this model of pregestational maternal diabetes indicate that reactive oxygen species (ROS) are critical players in mediating maladaptive heart development, leading to CHDs [13,14,15,16,17,18,19,20,21,22,23,24,25]. Oxidative stress, brought on by hyperglycemia, is not conducive to the development of cardiac progenitors as it results in the oxidation and inactivation of many molecules and enzymes necessary for cardiogenesis. Emerging data suggests a protective role of endothelial nitric oxide synthase (eNOS)-derived nitric oxide (NO) in reducing the incidence of the pathogenesis of diabetes-related CHDs. In this review, we examine the etiology and epidemiology of pregestational diabetes-induced CHDs and summarize numerous rodent studies looking at this link, while focusing on the pathophysiological role of ROS and eNOS uncoupling.

## 2. Congenital Heart Defects

### 2.1. Epidemiology and Classification

It is estimated that 257,000 Canadians and about 2 million Americans are living with a CHD [26,27]. Approximately 40,000 infants are diagnosed with a CHD each year in the US, and a quarter of these patients will need invasive treatment in the first year of life [3]. CHDs arise from perturbations in complex cellular and molecular processes underlying embryonic heart development. Adverse genetic and environmental factors can impede normal cardiogenesis and increase the likelihood of CHDs. Several types of malformations can result from missteps during this process. Defects of the septum, such as a ventricular septal defect (VSD) permits mixing of de-oxygenated and oxygenated blood. Major malformations such as transposition of the great arteries (TGA) can involve multiple structures of the heart [2]. Outflow tract defects such as persistent truncus arteriosus (PTA) and malformations in the valves of the heart can also arise from improper cardiogenesis. These defects compromise the function of the cardiovascular system by reducing the ability of the heart to adequately perfuse the body with oxygenated blood.

The International Classification of Diseases (ICD-10) classifies 25 distinct types of anatomical or hemodynamic CHDs [3]. CHDs can be categorized based on severity of the morphological lesions and functional consequences. Conventionally, CHDs are grouped into three categories: mild, moderate, and severe. The majority of CHD cases can be considered mild. The cardiac defects in these patients would be largely asymptomatic, often undergoing spontaneous resolution [28]. Moderate CHDs are more complex and require expert care, but are less intensive than the severe cases. Severe CHDs include the univentricular heart, heterotaxy, conotruncal defects, and atrioventricular canal defects, affecting one third of all patients with cardiac anomalies [29]. Expert cardiologic care and intervention are required for these patients in the first year of life. Accordingly, survival rate is dependent on the severity of the disease. This rate is estimated to be 98% for patients with a mild CHD, whereas it is only 56% for patients with a severe CHD [30]. Surgical procedures to correct congenital malformations of the heart do not always have high efficacy [31]. Individuals with CHDs commonly require multiple operations and drug therapy, and are at an elevated risk for arrhythmias, bacterial endocarditis, and heart failure, placing a large burden on the health care system [32]. Patients often have extracardiac co-morbidities, such as neurological deficits, which affect quality of life [33]. The prevalence of neurodevelopmental disabilities in the population of patients with CHDs ranges from 10% to over 50% depending on the severity of the lesion and whether pediatric surgery was required [34]. The spectrum of these abnormalities includes intellectual disability, autism spectrum disorder, and deficits in language, motor, and social skills [35]. In fact, attention deficit hyperactivity disorder is three to four times more prevalent in children with a CHD [36]. CHD-related hospitalizations in the US have steadily risen, with a financial burden of $6.1 billion per annum [37]. Therefore, it is vital that research be conducted into how these malformations develop to be able to identify potential preventative interventions.

### 2.2. Risk Factors Associated with CHDs

Risk factors associated with CHDs include genetic and nongenetic factors. While chromosome abnormalities and genetic mutations account for 15% of reported cases, nongenetic or environmental factors are associated with the majority of CHDs seen in the clinic [38]. Genetic factors include mutations in key transcription factors, such as *TBX5*, *NKX2.5*, that govern heart development, and chromosomal abnormalities, such as trisomy 21 [39,40]. Nongenetic factors are various environmental insults during pregnancy that compromise developmental pathways in the embryo. The nongenetic risk factors can include maternal diabetes, smoking, nicotine exposure, alcohol consumption, obesity, chlamydia or rubella infections, preeclampsia, and high altitude [41,42,43,44,45,46,47,48,49]. For instance, there was an exposure–response relationship between increasing body mass index in pregnant women and CHD incidence in their children. In fact, children of severely obese women were at a 1.94 times greater risk of having tetralogy of Fallot [44]. Pregestational diabetes is another risk factor for congenital heart disease [50], and has been observed to increase the risk of CHDs in human populations by more than four-fold [5,6,7,8,51,52,53].

## 3. Pregestational Diabetes and CHDs

### 3.1. Increasing Prevalence of Pregestational Diabetes

Currently, an estimated 451 million people are living with diabetes, and the incidence of this condition is increasing at a rate of 10 million people per year. It is projected that by the year 2040, approximately 693 million people will be affected [54]. In Canada, more than 11 million people are living with diabetes or prediabetes. Type 1 diabetes (T1D) is characterized by autoimmune destruction of pancreatic β-cells, resulting in a lack of insulin production; whereas type 2 diabetes (T2D) is the result of insulin resistance and usually has a later onset, but is much more prevalent [55]. The incidence of both T1D and T2D is increasing in youth aged 10–19 years [56,57]. In fact, T2D among the adolescent population in the US has increased 31% between 2001 and 2009, with prediabetes rising from 9% to 23% [3]. Importantly, 40% of women with diabetes are of reproductive age [9], and the incidence of T2D is rapidly increasing in this population [54]. It is estimated that 1 in 5 women are overweight/obese during pregnancy [58]. In the U.S alone, there was a 37% increase in the number of deliveries affected by pregestational or prepregnancy diabetes between 2000 and 2010 [59]. A study of trends in delivery hospitalizations between 1994 and 2004 found a diabetes diagnosis in 4.3 out of 100 deliveries, of which T2D saw the largest increase (367%) in diagnoses during this time period [60]. Other studies have shown an increase in pregnancies complicated with T1D and T2D by 44% and 90%, respectively in a 15 year period between 1998 and 2013 [61].

### 3.2. CHDs and Fetal Heart Abnormalities Associated with Pregestational Diabetes

A spectrum of cardiac defects is present in the offspring of women with pregestational diabetes, and ranges from minor structural malformations, such as atrial septal defect (ASD), to major morphological abnormalities, including tetralogy of Fallot [5,62,63]. In a national population-based pregnancy cohort in England, Wales, and Northern Ireland, similar rates of congenital malformations were reported between T1D and T2D, 4.8% and 4.3%, respectively [64]. In addition, it was found that 40% of infants born to diabetic mothers have hypertrophic cardiomyopathy, and case reports indicate that fetal death and stillbirth results from this condition in diabetic pregnancy [62,65,66,67,68]. It has also been demonstrated that diabetic pregnancy results in reduced fetal cardiac diastolic function [69], as well as altered heart rate variability and fetal acidaemia [70]. During development, a higher fetal heart rate has been reported by cardiotocographic analysis from the first to the third trimester in pregnancies complicated with pregestational diabetes [71,72]. Accordingly, offspring of these women also have an altered QRS complex [73]. In addition to modifying CHD risk and cardiac function, emerging longitudinal data suggests that adult metabolic and cardiovascular disorders are also in-part attributable to physiological insults in fetal life [74,75]. Adverse uterine environments, including maternal obesity, diabetes, undernutrition, and stress, can compromise embryonic development, reprogramming the epigenetic landscape [76]. These adverse outcomes are largely dependent on glycemic control during the first seven weeks of pregnancy, therefore, effective interventions must be able to protect this early developmental period.

The main focus of healthcare management is to maintain glycemic control, predominantly through the use of medications such as insulin and metformin. These drugs are not teratogenic, but their long-term effects on children’s health are not well understood. Current pharmacologic therapies have shortcomings that reduce their potential to prevent adverse fetal outcomes. For example, patient compliance and maintenance are major factors that can impact efficacy and glycaemic control in pregnant women with diabetes [77]. In addition, pregnancy can exacerbate and complicate diabetes management, usually requiring higher dosing of medication. As prevalence of diabetes is rapidly rising in low- to middle-income countries, there is also concern for adequate access to prenatal care and antidiabetic drugs [78]. Even with access to healthcare and medications, studies have found that glycemic control alone does not completely eliminate the risk for adverse events, and CHD incidence in the offspring of diabetic women has remained unchanged, showing little improvement [79,80,81,82]. Thus research must be conducted to find alternate interventions that support current treatment practices, while remaining safe and accessible. To this date, the underlying molecular mechanisms of diabetic teratogenesis are not fully understood. Signaling pathways such as Notch, Wnt, transforming growth factor-beta (TGFβ) and hypoxia-inducing factor (HIF1α) have been implicated in the pathogenesis of CHDs in maternal diabetes. Oxidative damage is largely believed to be at the root of cellular and molecular changes leading to the development of CHDs. Additionally, eNOS-derived NO is an important signaling molecule in early embryonic heart development. eNOS dysfunction and uncoupling also contribute to oxidative stress and CHD pathogenesis.

## 4. Embryonic Heart Development: Role of eNOS

### 4.1. Heart Morphogenesis

The heart is the first fully functional organ to form during embryogenesis, and its development is orchestrated by an interplay of conserved transcription factors that control growth, morphogenesis, and contractility [83]. Cardiac development involves the harmonic combination of three pools of progenitor cells, that when present in a specific spatial and temporal orientation together, form a self-excitable, four-chambered vascular pump [84]. Heart development commences at week 3 of human embryogenesis, corresponding to embryonic day 7.5 (E7.5) in the mouse, with the formation of a cardiac crescent and subsequent heart tube from cells of the anterior lateral mesoderm [85]. This first wave of cardiac progenitors has been termed the first heart field (FHF), forming a concentric inner layer of endocardial cells and outer layer of myocardial cells separated with extracellular matrix, also known as cardiac jelly [86]. The FHF-derived heart tube specifically contributes to parts of the atria and the left ventricle of the mature heart. The primitive heart tube is elongated at its poles by the addition of proliferative cells coming from the adjacent splanchnic pharyngeal mesoderm. This consecutive wave of myoblasts has been termed the second heart field (SHF), and contributes to the rightward looping morphogenesis of the tube, whereby primitive chambers of the heart can be distinguished [87]. These cells elongate the heart tube by migrating to its arterial and venous poles, progressively adding new myocardium and endocardium to the outflow and inflow tracts, respectively. Chamber development by ballooning, muscularization and fusion of endocardial cushions, ventricular myocardialization and septation, outflow tract and cardiac valve development are completed thereafter, with the addition of cardiac neural crest cells (CNCs) migrating from the dorsal neural tube [88]. Recently, single-cell RNA sequencing to map the transcriptome of the developing human heart revealed that cardiomyocytes exhibit differential gene expression signatures with chamber-specific clusters as early as 5 weeks into embryogenesis [89]. A number of conserved transcription factors govern the complex process of cardiac development. Mutations in *GATA4*, *NKX2.5* and *TBX5*, and *MEF2* are associated with CHDs in humans [90,91,92,93]. Specific spatiotemporal interaction and crosstalk between these factors are essential to control cardiogenic gene programs driving normal heart development [94]. Cardiac development is therefore a complex spatiotemporal process, and a misstep in this orchestration could lead to a CHD.

### 4.2. Nitric Oxide Signaling

NO was initially discovered in the vascular endothelium as an endothelium-derived relaxing factor (EDRF) by Furchgott and Zawadzki in 1980 [95]. NO is a lipophilic, small, gaseous molecule that has numerous roles in biological signaling processes and, most notably, serves as a potent vasodilator that regulates vascular function and blood pressure [96,97]. NO production within the cell is accomplished by the enzyme nitric oxide synthase (NOS), which catalyzes the conversion of L-arginine into nitric oxide through a NADPH-dependent reaction. Three isoforms of this enzyme, neuronal (nNOS, NOS1), cytokine-inducible (iNOS, NOS2), and endothelial (eNOS, NOS3) are distinctly expressed in mammalian cells and produce NO in a tissue-specific manner [98]. nNOS is constitutively expressed in neurons of the central and peripheral nervous system, whereas iNOS is expressed in cells of the immune system and produces high levels of nitric oxide associated with inflammatory processes. eNOS is membrane-bound and calcium-sensitive, expressed in endothelial and myocardial cells, located on chromosome 7 in humans [99]. They are homodimeric, haem-containing globular proteins, with an N-terminal oxygenase domain and a C-terminal reductase domain, which are linked by a calmodulin-binding sequence. The dimer produces NO in the coupled state and requires Ca^2+^/calmodulin, flavin adenine dinucleotide (FAD), flavin mononucleotide (FMN), and tetrahydrobiopterin (BH4) as cofactors [98]. Calcium-activated calmodulin initiates the election transfer to produce NO, which, when produced, acts on soluble guanylyl cyclase to catalyze the formation of the second messenger, cyclic guanosine monophosphate (cGMP). Downstream targets of cGMP include protein kinases (PKGs), ion channels, and cyclic nucleotide phosphodiesterases (PDEs) [100]. Activation or inhibition of protein function via S-nitrosylation is also a downstream effect of NO. Interestingly, eNOS can be itself S-nitrosylated, reducing its activity [101]. Phosphorylation of eNOS at serine-1177 by Akt (protein kinase B) activates the enzyme, which itself is active when phosphorylated. NO production via this PI3K/Akt/eNOS pathway represents a highly regulated biological process involved in endothelial cell permeability, vascular smooth muscle relaxation, calcium handling, immune regulation, and neurotransmission [97,102].

### 4.3. Tetrahydrobiopterin and eNOS Coupling

The pteridine (6R) 5,6,7,8-tetrahydrobiopterin (BH4) is an antioxidant and cofactor for many metabolic enzymes involved in the production of neurotransmitters and NO. As a cofactor, its heterocyclic ring structure functions to facilitate complex chemical reactions in a wide variety of biological processes [103]. BH4 is essential for NOS-mediated NO synthesis, by all three isoforms, and serves both biochemical and structural functions [103]. It is required for eNOS homo-dimer stabilization and is an allosteric modulator of l-arginine binding to the active site [104]. BH4 participates in the biochemical formation of NO from O_2_ by donating an electron for the conversion of l-arginine to l-citrulline [105]. It “couples” the heme reduction to NO synthesis and acts as a sequential one-electron reductant and oxidant [98].

BH4 production, either through de novo synthesis or recycling, promotes eNOS function, generating NO [106]. In fact, diminished levels of BH4 or depletion of l-arginine can uncouple the eNOS dimer, rendering it incapable of producing NO, instead generating oxygen-derived free radicals [107]. Similarly, in states of oxidative stress, this process is insulted at many levels, sustaining the oxidative environment and leading to cellular dysfunction. The production of ROS is often amplified because of a positive-feedback loop; ROS generates more ROS. For example, ROS causes the release of zinc from GTP cyclohydrase 1 (GTPCH-1), a rate-limiting enzyme in BH4 synthesis, rendering it less functional [108]. Consequently, the intracellular levels of BH4 decline, and eNOS is left without a cofactor, triggering its uncoupling and superoxide production. This superoxide anion participates in a rapid reaction with low levels of NO in the cell and forms the peroxynitrite anion (ONOO^−^) (Figure 1) [109]. This species is normally in equilibrium with peroxynitrous acid (ONOOH), which can decay and form ONOOH^•^, another reactive radical. Proteins are especially vulnerable to modification by the peroxynitrite system, as it incurs covalent changes to the amino acid residues cysteine, methionine, tryptophan, and tyrosine [110]. ONOO^−^ specifically inactivates GTPCH-1, decreasing de novo BH4 synthesis, and the additional oxidative stress furthers the oxidation of BH4 into BH2 [108]. BH2 is unable to couple eNOS, and competes with BH4 for eNOS binding, exacerbating the superoxide production. Mice lacking GTPCH-1 in the endothelium display significantly lower BH4 levels associated with eNOS uncoupling and increased ROS production, resulting in impairment of vasodilation [106].

Another enzyme inactivated by O_2_^•−^ and ONOO^−^ is dihydrofolate reductase (DHFR), responsible for the recycling of dihydrobiopterin (BH2) back to BH4 (Figure 1) [105,111]. Experiments done in *Escherichia Coli* (*E. coli*) demonstrate peroxynitrite-induced alterations to the catalytic site of DHFR, resulting in functional compromise [112]. Therefore, both the salvage and de novo synthesis pathways are crippled due to ROS, furthering the oxidative environment [113]. The relationship between eNOS function and diabetes has been well established. Endothelial dysfunction is a common consequence of diabetes and is mediated by oxidative stress-induced eNOS uncoupling [114,115,116]. Previous studies indicate that dissolution of the eNOS dimer also occurs in aging vessels and in cardiovascular disease states, such as hypertension, ischemia-reperfusion injury, and heart failure [117,118,119]. Treatment with BH4 has been shown to recouple eNOS and partially improve vascular endothelial function in diabetes [103,120]. Clinical studies also show the benefits of BH4 therapy in improving endothelial dysfunction, vasodilation, and lowering blood pressure in clinical settings of hypertension and hypercholesterolemia [121,122]. However, acute intracoronary infusion of BH4 to patients with coronary disease showed no significant improvement in coronary vascular reactivity or microvascular endothelial function in the heart [123]. Additionally, oral BH4 treatment for 2 to 6 weeks augments total biopterin levels in patients awaiting for coronary bypass surgery, but has no net effects on vascular redox state or endothelial function due to rapid oxidation of BH4 [124]. Thus, the type of cardiovascular disease that benefits from BH4 therapy remains to be investigated.

### 4.4. Role of eNOS in Heart Development

During embryogenesis, eNOS is expressed in the developing heart, and its expression has been shown to be vital for embryonic heart development [102,125]. In mice, expression of the synthase starts at E9.5 in the endothelium and cardiomyocyte precursors of the heart tube, and peaks at E13.5, after which it begins to decline, but remains detectable at birth and into postnatal development [102]. The lung, liver, gastrointestinal tract, reproductive organs, and brain all express eNOS during their organogenesis [126]. In vitro incubation of NOS inhibitors to embryonic stem cell-derived cardiomyocytes inhibited their maturation [126]. Interestingly, mouse embryonic stem cells exogenously treated with a NO donor or transfected with NOS promoted their differentiation into cardiomyocytes, inducing the expression of cardiac-specific genes [127]. Induced pluripotent stem cells and mouse E14.5 ventricular tissues in which eNOS are deficient show consistent transcriptome profiles, and particularly an up-regulation of glucose metabolism genes [128]. Gata4, a transcription factor needed for cardiomyocyte specification and heart development, can modulate and increase the expression of eNOS by binding to its promoter region [129]. The importance of eNOS in heart development has been showcased through the spectrum of cardiovascular anomalies seen in eNOS^−/−^ mice. Embryonic deletion of nNOS or iNOS displays normal cardiac phenotypes, however eNOS-null mice have a high rate of CHDs in the offspring. These mice have increased postnatal mortality, impaired heart function, and significant elevations in embryonic apoptosis in the heart [125]. In addition, they have valvular malformations, including bicuspid aortic valves (30–40% incidence) and underdeveloped mitral and tricuspid valves, which have significant regurgitation during systole along with aortic valve sclerosis and calcification [130,131,132].

During embryonic development, eNOS-null mice have less Snail^+^ mesenchymal cells in the endocardial cushions, suggesting impaired endocardial-to-mesenchymal transition (EndMT). This was coupled with significantly lower mRNA levels of genes involved in valve formation, such as *Tgf-β* and *Bmp2* [131]. Accordingly, patients with bicuspid aortic valve disease had lower eNOS protein expression in aortic endothelial cells compared with tricuspid aortic valves patient specimens [133]. eNOS-derived NO regulates cell growth and protects early cardiac progenitors against apoptosis [125]. This is accomplished through S-nitrosylation and inactivation of caspase-3 [134]. Additionally, shear stress-induced NO production and up-regulation of superoxide dismutase reduce the incidence of endothelial cell death via inhibiting caspase-3 activity [135]. eNOS is also required for cardiomyocyte proliferation, as well as VEGF expression to form the capillary network supplying the muscle [136]. In fact, eNOS intimately controls coronary artery development in mice, as loss of this enzyme results in coronary artery hypoplasia and spontaneous postnatal myocardial infarction [137]. Key transcription factors that control epithelial-to-mesenchymal transition (EMT) and vasculogenesis are downregulated in this model, including *Gata4, Wt1, VEGF, bFGF*, and erythropoietin, all of which are rescued by eNOS overexpression, restoring normal vasculature [137]. Overall, eNOS and NO are critical for the morphogenesis of all major components of the developing heart, and research characterizing their functions has been previously summarized by our group in a more extensive review [102].

## 5. Oxidative Stress and Diabetes

### 5.1. The Anatomy of Reactive Oxygen Species

ROS can exist as neutral molecules, ions, or radicals, such as hydrogen peroxide (H_2_O_2_), the superoxide anion (O_2_^•−^), or the hydroxyl radical (^•^OH), respectively [138]. They are produced in subcellular compartments and microdomains as part of cellular metabolic activity. Their biological effects, physiological or pathological, depend on their spatiotemporal characteristics, including the local levels, duration of their release, and interactions between ROS in different subcellular compartments [138,139]. Superoxide is considered a primary ROS and can either be generated from a metabolic process or by activation of oxygen by irradiation. Superoxide cannot readily cross plasma membranes, therefore having local, transient effects. The superoxide anion can react with nitric oxide, forming the highly reactive peroxynitrite [140]. This anion can also be dismutated into hydrogen peroxide (H_2_O_2_) by superoxide dismutase (SOD). Notably, H_2_O_2_ is diffusible across membranes via peroxiporins, a type of aquaporin, to achieve its biological effects in different subcellular compartments [141,142]. While basal physiological level of H_2_O_2_ is required for redox signaling, excess production of H_2_O_2_ in the cell is detrimental as it can easily be converted into hydroxyl radical ^•^OH [143,144]. H_2_O_2_ also serves as an endothelium-derived hyperpolarizing factor (EDHF), mediates flow-induced dilation in human coronary arterioles, and contributes to the maintenance of vascular homeostasis, along with other vasodilating factors such as prostacyclin, and NO [145]. In the endothelium, H_2_O_2_ is produced from dismutation of superoxide anion generated from several sources including mitochondrial electron transport chain, NADPH oxidase, xanthine oxidase, and importantly eNOS. Thus, eNOS in the endothelium produces not only NO (more pronounced in large conduit vessels), but also H_2_O_2_ (predominantly in resistance arteries) to regulate vascular function [145].

Closely linked with the generation of free radicals are redox-active metals. An iron redox couple is associated with the redox state of the cell, and is kept under strict physiological control. In conditions of excessive stress, and excess O_2_^•−^ release, free iron released from enzymes takes part in Fenton’s reaction and generates secondary ROS, hydroxyl radicals (Figure 2) [146]. Elevated concentrations of ROS can react with proteins, lipids, carbohydrates, and nucleic acids causing nonspecific damage and permanent functional changes. Hydroxyl radicals produced in close proximity to DNA or RNA can react, resulting in mutations or modifications [147]. Over 20 different types of RNA base damage have been identified, 8-hydroxyguanosine (8-OHG) being the most prevalent oxidized base (Figure 2) [148]. The hydroxyl radical reacts with the guanine to form C8-OH adduct radical, as it is highly reactive. Next, 8-OHG is generated from the loss of an electron (e^−^) and proton (H^+^) [148]. The hydroxyl radicals can also react with polyunsaturated fatty acid residues of phospholipids, the main component of the cell membrane, which are highly sensitive to oxidation to result in lipid peroxidation (Figure 2) [149]. Along with mitochondrial ROS production, intracellular sources of ROS include peroxisomes, lysosomes, and enzymes such as NADPH oxidase, xanthine oxidase, and cytochrome p450 [138].

### 5.2. Oxidative Stress Induced by Hyperglycemia

Maternal health and environmental factors have both short-term and long-term consequences for the offspring [150]. Hyperglycemia in the mother alters the environment in which embryos develop because glucose, among other metabolic molecules, can pass freely through the placental barrier. In this altered fetal environment, oxidative stress can result from multiple pathways (Figure 3). Excess glucose can be freely taken up by embryonic heart through glucose transporter 1 (GLUT1), the predominant GLUT transporter in the developing heart [151]. The developing heart is sensitive to changes in glucose availability, and early cardiac progenitors mainly rely on glycolysis for energy production. In fact, GLUT1 expression is upregulated in the embryonic heart in experimental models of maternal diabetes, facilitating the glucose overload [76]. However, as cardiomyocytes terminally differentiate they switch to mitochondrial oxidative metabolism pathways [152], and the insulin-sensitive glucose transporter GLUT4 becomes the predominant form postnatally [153].

High levels of glucose promote glycolysis, and electron carriers such as NADH and FADH_2_ are increasingly produced, depleted, and reproduced through cycles of glycolysis and oxidative phosphorylation within the mitochondria (Figure 3). These molecules act as electron sources for the mitochondrial electron transport chain, and higher levels of electrons flowing through this chain will lead to increased oxygen consumption and ATP formation [154]. At several points during the oxidative phosphorylation process, electrons can react with oxygen, forming radicals [140]. The end result of high glucose is an overproduction of electron donors and generation of reactive oxygen species (ROS) such as superoxide and hydrogen peroxide [154]. This notion is supported by experiments showing that overexpression of uncoupling protein-1 (UCP-1) abolished the proton electrochemical gradient driving oxidative phosphorylation and prevented hyperglycemia-induced ROS production [155].

Moreover, mitochondrial ROS can stimulate the production of cytosolic ROS through several mechanisms. Hydrogen peroxide can lead to the activation of cellular Src kinase, which promotes NADPH oxidase activity, producing superoxide [156]. These reactive molecules can damage DNA and proteins critical for organogenesis [157]. Usually, under conditions of normal NADH to NAD^+^ ratio, the production of ROS is relatively low and is reduced by endogenous antioxidants such as superoxide dismutase (SOD), catalase, and glutathione peroxidase [158,159]. High-glucose environments during development disrupt intracellular redox homeostasis by stimulating ROS production, to exceed levels that can appropriately be managed by endogenous antioxidants. Oxidative stress is also potentiated by attenuated glutathione synthesis in diabetic conditions [160]. Further, increased abundance of glucose and glycolysis intermediates stimulate other metabolic pathways that can lead to cellular damage. For example, the hexosamine pathway produces UDP-N-acetylglucosamine (UDP-GlcNAc) from fructose-6-phosphate [161]. This molecule is involved in *O*-linked N-acetylglucosamine modification of proteins, inhibiting their functional capacity. Furthermore, advanced glycation end-products (AGEs) are produced by the advanced glycation and oxidation of proteins, lipids, and nucleic acids, and are thought to play a role in the development of diabetic complications. AGEs activate RAGE (the receptors for AGEs) signaling and cause carbonyl stress, leading to activation of NF-kB, inflammatory response, ROS production, and endothelial dysfunction [162,163]. Overall, the abundance of ROS and the depletion of antioxidants lead to lipid peroxidation, as well as damage to nucleic acids and proteins [164]. In diabetic pregnancies, eNOS dysfunction and oxidative stress/damage ultimately alters gene expression, cell proliferation, survival, and inhibits EMT, resulting in CHDs in the offspring (Figure 3).

## 6. Involvement of eNOS Uncoupling and ROS in Pregestational Diabetes-Induced CHDs

### 6.1. Experimental Models of CHDs Induced by Hyperglycemia and Pregestational Diabetes

Hyperglycemia has been recognized as a teratogen causing embryopathy associated with neural tube defects and CHDs [165,166]. To simulate CHDs in humans due to pregestational diabetes, both ex vivo and in vivo animal models have been employed. Whole embryo culture of rodents allows direct observation of the growth and development of embryos outside the maternal uterus for up to 5 days. Additionally, culture conditions can be easily manipulated, for example using high glucose concentrations to mimic hyperglycemia. Although the rodent whole embryo culture was initially described about 60 years ago, because of these advantages, it is still being used to study organogenesis [167]. Chick embryo culture has also been used to study the effects of hyperglycemia on heart development. Exposure of neural crest cells to high glucose in vivo results in double-outlet right ventricle (DORV) and ventricular septal defect (VSD) [166]. However, the in vivo rodent models induced by either streptozotocin (STZ) with various administration regimens or high-fat diet plus low-dose STZ to simulate type 1 or 2 diabetes, respectively, are the most commonly used and clinically relevant models of maternal diabetes. We have employed a mouse model of pregestational diabetes induced by STZ, which produced a wide spectrum of CHDs in the offspring, including ASD, VSD, atrioventricular septal defect (ASVD), DORV, tetralogy of Fallot, transposition of great arteries (TGA), and hypoplastic left heart syndrome (HLHS) [15,16,17,18]. Some of the fetuses exhibit coronary artery malformations with reduced coronary artery diameter and abundance, resembling hypoplastic coronary artery disease (HCAD) in humans [16,17]. Studies using these in vivo rodent models to examine the roles of NO and ROS are summarized in Table 1.

### 6.2. eNOS Uncoupling and CHDs in Pregestational Diabetes

The molecular mechanisms by which hyperglycemia causes CHDs are complex and may involve multiple signaling pathways. Interestingly, eNOS expression and NO levels are reduced in fetal hearts of offspring of diabetic dams [13]. The dysfunctional eNOS induced by high glucose stress is a result of reduced chromatin accessibility at the *Nos3* locus and decreased eNOS phosphorylation of serine 1177 by the protein kinase Akt [14,174]. Furthermore, lower NO levels are associated with higher Jarid2 expression and its enhanced interaction with the *Notch1* locus, leading to decreased Notch1 expression and a higher incidence of VSDs in offspring of maternal diabetes [14]. Notably, pregestational diabetes lowers fetal BH4 levels, and induces eNOS uncoupling, elevated superoxide generation, as well as altered cardiac gene expression of *Gata4*, *Gata5*, *Nkx2.5*, *Tbx5*, and *Bmp10* in the fetal heart [15]. Moreover, mRNA levels of GTPCH1 and DHFR, enzymes involved in BH4 synthesis, were also significantly reduced at E12.5 in the fetal heart of offspring from diabetic dams, compared with control [15]. To elucidate the role of eNOS uncoupling in the pathogenesis of CHDs, diabetic dams were treated with sapropterin (Kuvan^®^), an orally active synthetic form of BH4. Our results showed that oral sapropterin treatment in the diabetic dams recouples eNOS and lowers ROS levels in fetal hearts. Further, impaired cardiac gene expression and cell proliferation induced by maternal diabetes were normalized with sapropterin treatment. Most strikingly, the incidence of CHDs was lowered from 59% to 27%, and major abnormalities such as AVSD and DORV were absent in the sapropterin-treated group [15]. The study shows a pivotal role of eNOS uncoupling in the pathogenesis of CHDs in diabetic pregnancies (Table 1).

### 6.3. Role of ROS in Pathogenesis of CHDs in Pregestational Diabetes

Elevated ROS levels and oxidative damage have been proposed to contribute to diabetic complications [165,175]. Ex vivo cultures of rat embryos under high-glucose conditions show glutathione depletion, excessive ROS production, growth retardation, and severe malformations after exposure to high glucose [176,177]. In a U-substrain of Sprague-Dawley rats at Uppsala University, Sweden, pregestational diabetes induced by STZ resulted in craniofacial and skeletal malformations in the offspring [165]. Additionally, the rat offspring of maternal diabetes also show a variety of CHDs including DORV, persistent truncus arteriosus, VSD, teratology of Fallot, and pharyngeal arch artery defects [19,168]. Furthermore, maternal treatment with antioxidants such as butylated hydroxytoluene and vitamin C and E reduced the rate of craniofacial and skeletal malformations, while treatment with vitamin E decreased the rate and severity of CHDs in rats, suggesting the involvement of ROS [19,178,179,180]. Microarray analysis shows that genes involved in apoptosis, proliferation, migration, and differentiation in fetal hearts at E13.5 and E15.5 are differentially expressed in embryos of diabetic pregnancy [169]. These changes in gene expression and downregulation of Pax3 are associated with cardiac malformation such as persistent truncus arteriosus and VSD [170]. Using reporter mice linked to Pax3 expression, defective cardiac neural crest cell migration and outflow tract defects were found in embryos of maternal diabetes/hyperglycemia, and reduced by antioxidant treatment with glutathione ethyl ester or vitamin E [20].

Subsequent studies from us and others demonstrated that both ROS and oxidative stress are elevated in the fetal heart of maternal diabetes. Notably, inhibition of ROS using oral treatment of antioxidant N-acetylcysteine or by transgenic overexpression of superoxide dismutase 1 (SOD1) lowers fetal heart ROS levels and reduces the incidence of CHDs and coronary artery malformations induced by pregestational diabetes [17,18,21,22]. Interestingly, intravenous administration of miRNA-containing exosomes isolated from diabetic dams to control dams at E8.5 or E11.5 is sufficient to induce VSDs in fetuses [171]. Further, expression of genes and miRNAs critical to heart morphogenesis is regulated by ROS in the fetal heart of maternal diabetes since the changes were normalized by N-acetylcysteine treatment or overexpression of SOD1 [17,18,23]. Additionally, cell apoptosis and endoplasmic reticulum (ER) stress are involved in CHD pathogenesis in maternal diabetes as deficiency of apoptosis signal-regulating kinase 1 (ASK1) lowers apoptosis, ER stress, and incidence of CHDs in offspring of pregestational diabetes [24]. Together, these studies support a critical role of ROS in the pathogenesis of CHDs in pregestational diabetes in rodents (Table 1).

It should be noted that maternal supplementation with antioxidants vitamin E and C during pregnancy in general population shows no overall benefit or harm for the prevention of stillbirth, neonatal death, poor fetal growth, preterm birth, or pre-eclampsia [181,182]. In fact, some studies show that high maternal vitamin E intake by diet or supplements even increases the risk of CHDs in children [183]. These results are consistent with a recent meta-analysis showing no effect of vitamin E and C supplementation on the cardiovascular outcomes including myocardial infarction, stroke, total death, and cardiac death in adults [184]. The failure to translate preclinical paradigms into clinical settings underscores our limited knowledge on ROS biology, particularly the subcellular compartmentation of ROS sources and antioxidant systems, which may dictate their final effect. However, whether N-acetylcysteine or BH4 affects the incidence of CHDs of children in pregestational diabetes patients remains to be investigated.

Additionally, AGEs represent another mechanism implicated in hyperglycemia-induced malformations. Specifically in the rat embryo, 3-deoxyglucasone, a precursor in the formation of AGEs was demonstrated to be increased in high-glucose compared with low-glucose conditions. The increased level of 3-deoxyglucasone was associated with an increased rate of embryonic malformations [185]. Furthermore, maternal diabetes in rats increased the accumulation of AGEs such as Nε(carboxymethyl)lysine (CML) in areas susceptible to diabetes-induced CHDs, including the outflow tract of the heart and the aortic arch [186]. These studies suggest that AGEs are associated with congenital malformations, although their definitive role in the pathogenesis of CHDs remains to be determined.

## 7. Maternal Exercise Reduces Incidence of CHDs from Pregestational Diabetes

Clinical studies suggest an overall healthy lifestyle with regular exercise before and during pregnancy is important to ensure a healthy outcome of the newborns [187]. Exercise has a number of benefits including, but not limited to, improvements of metabolism, cardiovascular health, cognition, fertility, bone health, immune response, and even slowing down aging [188]. Exercise during gestation lowers the risk of gestational diabetes mellitus and improves offspring health [189]. A joint Society of Obstetricians and Gynaecologists of Canada (SOGC)/Canadian Society for Exercise Physiology (CSEP) clinical practice guideline recommends 150 min of moderate-intensity exercise per week during pregnancy to achieve clinically meaningful health benefits and reductions in pregnancy complications [190]. Remarkably, maternal voluntary exercise for 3 months before conception has recently been shown to lower the incidence of VSDs in offspring of old mothers, suggesting maternal exercise reduces the risk of CHDs associated with maternal age [191]. Using a STZ diabetes mouse model, we examined the effects of maternal exercise on pathogenesis of CHDs induced by pregestational diabetes [16]. Our results show that maternal exercise reduces ROS levels and oxidative stress and improves eNOS phosphorylation in the fetal hearts of offspring from mice with diabetes. Most notably, maternal exercise lowered the incidence of CHDs and coronary artery malformation by 58% [16]. Few other studies have examined changes in heart development associated with maternal exercise. There is evidence that exercise during pregnancy may influence the maturation of cardiac autonomic control and increase right ventricular cardiac output in children [192,193], and improve NO bioavailability and vascular function in the fetus of pigs [194].

Although several exercise-mediated adaptations have been identified that may play a role in modulating cardioprotective response, the mechanism by which maternal exercise alters fetal heart development remains unclear. Exercise has been demonstrated to influence gene expression and activate pathways that alter angiogenesis, proliferation, and cellular metabolism [188]. Peroxisome proliferator-activated receptors-gamma (PPARγ) coactivator-1a (PGC-1α) is a master regulator of energy metabolism. In the heart, PGC-1α is the major coactivator for transcription factors PPARα and PPARγ, which regulate cardiac metabolism [195]. The PPARβ/δ isoform has been demonstrated to protect cardiomyocytes against apoptosis induced by oxidative stress, and is known to be induced by exercise in skeletal muscle [196]. Through activation of other transcription factors, PGC-1α also stimulates mitochondrial biogenesis and promotes the expression of electron transport chain proteins. PGC-1α is upregulated with exercise and has been shown to regulate mitochondrial antioxidant defense in vascular endothelial cells [196]. Laker et al. found that maternal exercise modified the methylation status of the PGC-1α promoter and increased its gene expression in the offspring [197]. In future studies, it would be worthwhile to evaluate the involvement of PGC-1α in mediating the effects observed in our model of pregestational diabetes.

In adults, exercise elevates ROS production, which paradoxically leads to improved redox capacity through upregulation of antioxidant enzymes in cardiac and skeletal muscle [198,199]. In fact, exercise-induced resistance to ischemia-reperfusion injury is mediated by such adaptations to antioxidant defenses [200]. Promotion of NO signaling through exercise is also known to contribute to this protective mechanism [201]. During exercise, the β_3_-adrenergic receptor is stimulated, and its activation increases eNOS activity and NO bioavailability [202]. Activation and phosphorylation of eNOS-Ser1177 in the heart is mediated by Akt activation. As Akt activation has been shown to promote cardiomyocyte proliferation and expansion of cardiac progenitors [203], this pathway may also have contributed to protection against CHD pathogenesis observed in our model [16]. As of now, fetal adaptations in response to maternal exercise has not been well characterized. Our results suggest that maternal exercise may improve the capacity to maintain redox balance in the fetal heart. Results from recent studies support this notion, showing that maternal exercise resulted in increased eNOS and extracellular SOD expression in vascular smooth muscle cells, increased mitochondrial enzymatic activity, and decreased ROS in fetal cardiovascular tissues [204,205].

All in all, maternal exercise may protect against abnormal heart development through multiple mechanisms. For example, exercise has been shown to decrease the expression of transcription factor C/EBPβ in the heart, leading to the upregulation of cardiac factors *Gata4*, *Tbx5*, and *Nkx2.5* to promote cell proliferation, differentiation, and physiological hypertrophy in adults [206]. There is also emerging interest in epigenetic regulation of gene expression and micro-RNAs (miRNA) involved in heart development. Specifically, miR-1/miR-133 clusters in the heart are regulated by myocyte-enhancer factor 2 (Mef2) and Nkx2.5 and are involved in regulation of cell proliferation, apoptosis, and differentiation during heart development. Some of these miRNAs are upregulated with exercise [207]. The exact role of eNOS and ROS signaling in the exercise-induced benefits remains to be determined, and there are many exciting directions for possible future studies.

## 8. Conclusions

It is clear that the diabetes epidemic poses a major threat to both high- and low-income countries alike, given the alarming rate at which prevalence is rising globally [54]. CHDs are an important consequence of pregestational diabetes, causing significant morbidity, mortality, and burden on healthcare costs. The teratogenic potential of maternal hyperglycemia has been well documented for generations, and research thus far has uncovered perturbations to ROS and NO homeostasis to be major drivers of this phenomenon. Cardiogenesis is a highly complex process governed by a multitude of factors that are sensitive to oxidative damage, including eNOS, the enzyme that produces NO. Unfortunately, clinical evidence suggests that insulin treatment and glycemic control alone does not completely eliminate the risk of CHDs in the children of women with pregestational diabetes. Animal studies have identified several antioxidants to significantly protect against CHDs induced by maternal diabetes. Factors such as BH4, which bridges the homeostasis of both ROS and NO, show promise to be potential treatment in the prevention of CHDs during maternal diabetes. Additionally, lifestyle modifications such as maternal exercise, which reduces ROS and increases eNOS activity, may be a simple yet effective intervention to supplement pharmacological therapy and minimize risks of birth defects in mothers with diabetes. In conclusion, although great strides have been made in understanding the pathogenesis of CHDs, a significant paucity in human data exists, limiting the advancement of clinical treatment. Moving forward, further translational research is needed in order to evaluate the efficacy of such interventions in human populations.

## Figures and Tables

**Figure 1 antioxidants-08-00436-f001:**
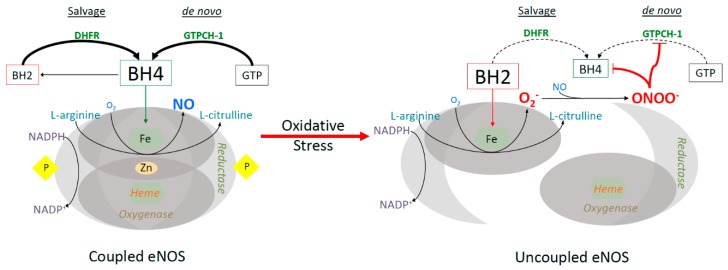
Endothelial nitric oxide (eNOS) uncoupling in states of oxidative stress. The functional eNOS dimer, producing nitric oxide, is seen on the left. eNOS dysfunction, on the right, occurs in states of oxidative stress, where BH4 is oxidized, eNOS is undimerized and uncoupled. In this state, nitric oxide is no longer produced and instead superoxide radical is generated. The salvage and de novo BH4 biosynthesis pathways are actively producing BH4 in normal conditions, but are impaired by oxidative stress, specifically derived from eNOS uncoupling.

**Figure 2 antioxidants-08-00436-f002:**
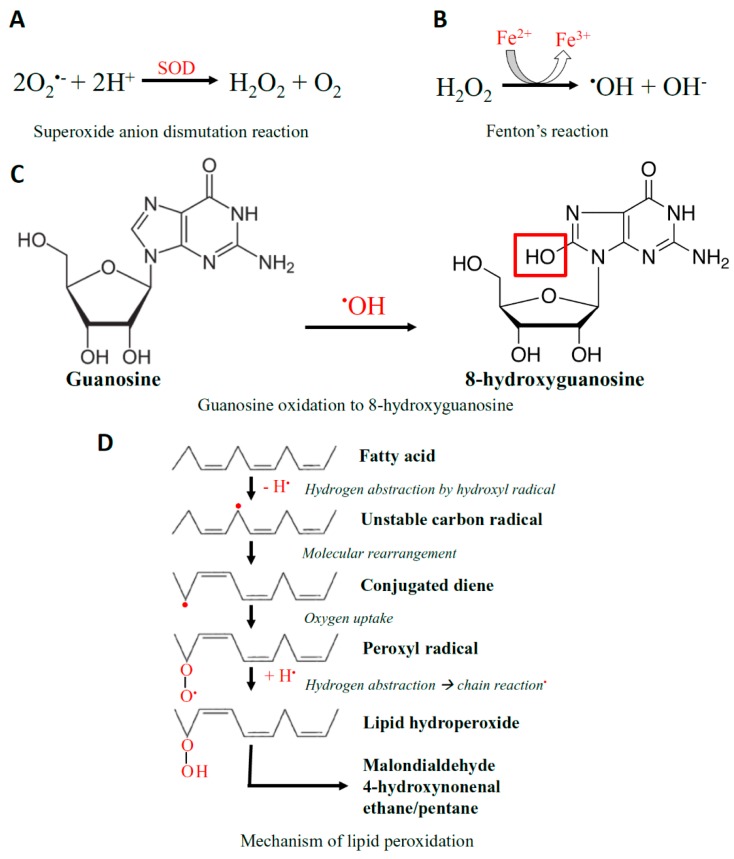
The Chemistry of ROS. (**A**) Superoxide anion (O_2_^•−^) undergoes a dismutation reaction in the presence of the superoxide dismutase (SOD) enzyme, converting it into hydrogen peroxide (H_2_O_2_) and oxygen. (**B**) H_2_O_2_ is then converted to secondary ROS, being hydroxyl radical (^•^OH) by Fenton’s reaction. (**C**) Hydroxyl radicals are capable of promoting the oxidation of guanosine to 8-hydroxyguanosine. (**D**) The hydroxyl radical is also able to react with polyunsaturated fatty acid residues of phospholipids, which are highly sensitive to oxidation, resulting in lipid peroxidation.

**Figure 3 antioxidants-08-00436-f003:**
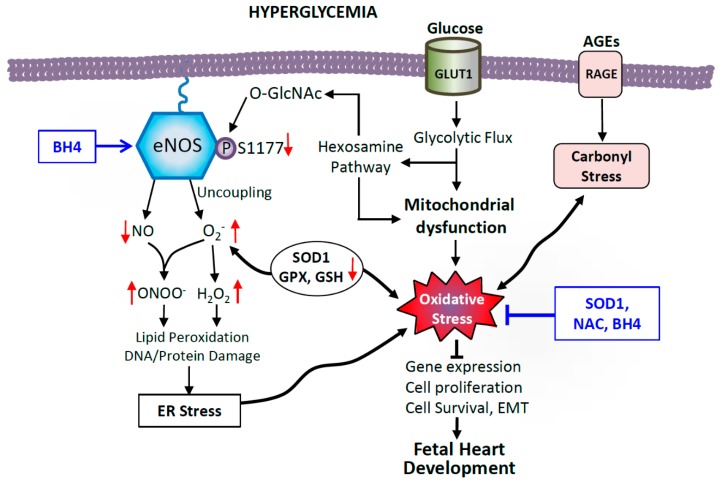
Possible mechanisms of hyperglycemia-induced cellular oxidative damage in the fetal heart. Excess glucose increases the rate of glycolytic metabolic pathways, overproducing electron donors that increase mitochondrial reactive oxygen species (ROS) production. Coupled with impaired antioxidant defenses including lower levels of superoxide dismutase 1 (SOD), glutathione peroxidase (GPX) and reduced glutathione (GSH), this leads to oxidative stress in diabetes. Glycolysis intermediate fructose-6-phosphate also feeds into the hexosamine pathway to produce UDP-N-acetylglucosamine (UDP-GlcNAc), which leads to impairment of endothelial nitric oxide synthase (eNOS) function by O-GlcNAc modification of eNOS at serine 1177, the site of phosphorylation by Akt activation. Oxidative stress also causes eNOS to undimerize and produce superoxide, which reacts with NO to potentiate oxidative and nitrosative stress. Additionally, advanced glycation end-products (AGEs), though binding to receptors of AGEs (RAGE) induce carbonyl stress and promote ROS. Dysfunctional eNOS and oxidative stress alter gene expression, cell proliferation, survival, and inhibit epithelial-to-mesenchymal transition (EMT), leading to CHDs, which can be ameliorated by SOD1 overexpression, and maternal treatment with N-acetylcysteine (NAC) and sapropterin (BH4).

**Table 1 antioxidants-08-00436-t001:** ROS and eNOS involvement in rodent studies on pregestational diabetes-induced congenital heart defects.

Rodent Strain/Type of Diabetes	Treatment/Genetic Modification	eNOS/ROS Involvement	Altered Fetal Heart Signaling Pathways/Cellular Processes by Maternal Diabetes	Spectrum of CHDs in Offspring	References
Swiss albino mice STZ 75 ^mg^/_kg_ IP × 3	N/A	eNOS and ROS	↓ eNOS, ↑ VEGF ↓ NO release, ↑ apoptosis, cell blebbing, and aggregation *	N/A	[13]
C57BL/6 STZ 75 ^mg^/_kg_ IP × 3	N/A	eNOS and ROS	↑ ROS, ↓ Notch1/eNOS pathway	VSD	[14]
C57BL/6 STZ 50 ^mg^/_kg_ IP × 5	Sapropterin	eNOS and ROS	eNOS uncoupling ↓ proliferation, ↓ SHF contribution, ↓ cardiac valve remodeling	ASD, VSD, AVSD, DORV, PTA, Valve Thickening, Thin ventricles	[15]
	Exercise	eNOS and ROS	↓ eNOS phosphorylation ↓ proliferation, ↓ Wt1 expression	ASD, VSD, AVSD, DORV, HLHS, HRHS, Coronary artery malformation	[16]
C57BL/6 STZ 75 ^mg^/_kg_ IP × 3	N-acetylcysteine	ROS	↓ proliferation, ↓ Epicardial EMT	Coronary artery malformation	[17]
			↓ proliferation, ↑ apoptosis	ASD, VSD, AVSD, DORV, TGA, TOF	[18]
SD Rats STZ 40 ^mg^/_kg_ IV × 1	Vitamin E	ROS	N/A	VSD, DORV, PTA, PDA	[19]
FVB STZ 100 ^mg^/_kg_ IV × 1	Vitamin E and Glutathione	ROS	↑ apoptosis, ↓ CNC migration	PTA	[20]
C57BL/6 STZ 75 ^mg^/_kg_ IP × 2	SOD-1 overexpression	ROS	↓ Wnt/β-catenin signaling ↓ proliferation, ↑ apoptosis	VSD	[21]
		ROS	↓ TGFb/Smad signaling ↓ proliferation **	N/A	[22]
		ROS	↓ proliferation and ↑ apoptosis ^†^	N/A	[23]
	Ask1 knockout	ROS	↑ ASK1/JNK signaling ↑ ER stress, ↓proliferation, ↑ apoptosis	VSD, PTA, OFT defects	[24]
SD Rats, HFD + STZ 65 ^mg^/_kg_ IP × 1	N/A	ROS	↑ lipid droplets, abnormal mitochondrial structure & membrane potential	N/A	[25]
SD Rats STZ 40 ^mg^/_kg_ IV × 1	N/A	N/A	↑ apoptosis	VSD, AVSD, DORV, DILV	[168]
Swiss albino mice STZ 75 ^mg^/_kg_ IP × 3	N/A	N/A	↓ proliferation, ↑ apoptosis, ↓ migration, ↓ differentiation, ↑ O-glycan biosynthesis, ↓ lysosome activity ^†^	N/A	[169]
	N/A	N/A	↓ proliferation, ↑ apoptosis, swollen mitochondria, ↓ myofilaments, ↓ adherence junctions	VSD, PTA	[170]
C57BL/6 STZ 50 ^mg^/_kg_ IP × 5	N/A	N/A	N/A	VSD	[171]
SD Rats STZ 35 ^mg^/_kg_ IP × 1	N/A	N/A	↑ apoptosis, ↑ mitosis	↑ heart size, ↓ cardiac function	[172]
FVB Mice STZ 100 ^mg^/_kg_ IP × 2	N/A	N/A	↑ Hif1α/VEGF signaling ↓ proliferation, ↑ apoptosis	VSD, ↓ myocardial volume	[173]

**Abbreviations:** N/A, not available; IP, Intraperitoneal; IV, intravenous; SD, Sprague-Dawley; STZ, Streptozotocin; HFD, high fat diet; SHF, Second heart field; CNC, Cardiac neural crest; ASD, atrial septal defect; VSD, ventricular septal defect; AVSD, Atrioventricular septal defect; PTA, persistent truncus arteriosus; PDA, patent ductus arteriosus; DORV, double-outlet left ventricle; DILV, double-inlet left ventricle; OFT, outflow tract; TOF, tetralogy of Fallot; HLHS and HRHS, hypoplastic left and right heart syndrome, respectively; ↑ indicates an elevation, ↓ indicates a reduction. * in vitro, ** ex vivo, ^†^ gene ontology analysis.

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
