# Peer review of "Say NO to ROS: Their Roles in Embryonic Heart Development and Pathogenesis of Congenital Heart Defects in Maternal Diabetes"

_antioxidants, 2019, doi:10.3390/antiox8100436_

Round 1

Reviewer 1 Report

The paper broadly reviews the epidemiology of congenital heart disease in relation to gestational risks (e.g. diabetes), and specifically examines the relationship to ROS and dysfunctional eNOS. 

The following points should be considered;

i. on lines 259, 415, the authors mention some animal studies using BH4 supplementation leading to NOS re-coupling and decreased ROS. They should acknowledge that this approach has not consistently been effective in other models or in vivo studies with eNOS uncoupling and vascular disease. In fact, in many cases, the BH4/BH2 ratio was not improved, due to rapid oxidation of BH4. 

ii. in the same line of thinking, the authors should more cautiously present the evidence on the use of systemic antioxidants; while some of them may have worked in the specific gestational (pre-clinical) models presented, in general, this approach has given disappointing results in all clinical studies on cardiovascular diseases associated with similar risk factors and increased oxidant stress. The text, as formulated, would give the reader a misleading impression that antioxidants are ripe for clinical use to prevent congenital heart disease in pregnancies at risk.

iii. the failure to translate pre-clinical paradigms into new treatments probably stems from a still incomplete understanding of the biology of intracellular ROS. The paragraphs on ROS should highlight such gaps in our knowledge, particularly on the sub-cellular compartmentation of ROS sources and antioxidant systems, that probably dictate their final effect. In the text, these concepts are generally overlooked and the authors do not take more recent evidence into account. For example, on line 302, they state that "H2O2 is freely diffusible". This conventional view has been put in question and is not compatible with recent evidence for the dual role of H2O2 as mediator of both physiological signaling and pathological damage. (see for example, Redox Biol. 2017 Apr;11:613-619 or Free Radic Biol Med. 2016 May;94:157-60). Similar concepts on NOS are discussed in recent reviews (Nat Rev Cardiol. 2018 May;15(5):292-316).

iv. the same over-simplification is apparent in the paragraphs on exercise. There is no doubt about its benefits, including in maternal diabetes, but whether the benefit can be reduced to antioxidant effects is doubtful, particularly because other evidence suggests that ROS mediate the positive effects of exercise in other settings.  Again, more elaboration is needed there.

v. some typos: l. 151 "rising" should be read; Table 1: some characters misplaced in the columns

Author Response

We thank you for your insightful and constructive comments to improve our manuscript.

i. on lines 259, 415, the authors mention some animal studies using BH4 supplementation leading to NOS re-coupling and decreased ROS. They should acknowledge that this approach has not consistently been effective in other models or in vivo studies with eNOS uncoupling and vascular disease. In fact, in many cases, the BH4/BH2 ratio was not improved, due to rapid oxidation of BH4.

Response: We agree with your comments and have now included studies that did not show a beneficial effect of BH4 treatment. Rapid oxidation of BH4 as a possible reason for the negative effects of BH4 is also included. Please see line 264-272.

ii. in the same line of thinking, the authors should more cautiously present the evidence on the use of systemic antioxidants; while some of them may have worked in the specific gestational (pre-clinical) models presented, in general, this approach has given disappointing results in all clinical studies on cardiovascular diseases associated with similar risk factors and increased oxidant stress. The text, as formulated, would give the reader a misleading impression that antioxidants are ripe for clinical use to prevent congenital heart disease in pregnancies at risk.

Response: Your comments are well taken. Accordingly, we have included clinical studies that did not show an overall benefit or harm of maternal supplementation with vitamin E and C during pregnancy on maternal and fetal outcomes or even increased the risk of CHDs in children. Please see text on line 492-511.

iii. the failure to translate pre-clinical paradigms into new treatments probably stems from a still incomplete understanding of the biology of intracellular ROS. The paragraphs on ROS should highlight such gaps in our knowledge, particularly on the sub-cellular compartmentation of ROS sources and antioxidant systems, that probably dictate their final effect. In the text, these concepts are generally overlooked and the authors do not take more recent evidence into account. For example, on line 302, they state that "H2O2 is freely diffusible". This conventional view has been put in question and is not compatible with recent evidence for the dual role of H2O2 as mediator of both physiological signaling and pathological damage. (see for example, Redox Biol. 2017 Apr;11:613-619 or Free Radic Biol Med. 2016 May;94:157-60). Similar concepts on NOS are discussed in recent reviews (Nat Rev Cardiol. 2018 May;15(5):292-316).

Response: Thank you for your insightful comments. Accordingly, our limited knowledge on subcellular compartmentation of ROS is now highlighted on lines 314-318. Also, diffusion of H2O2 across membranes via peroxiporins is now emphasized (lines 322-324). The dual role of H2O2 as mediator of both physiological signaling (e.g., EDHF) and pathological damage is outlined (see lines 324-333). Suggested references are now cited (refs. 96, 141 and 144).

iv. the same over-simplification is apparent in the paragraphs on exercise. There is no doubt about its benefits, including in maternal diabetes, but whether the benefit can be reduced to antioxidant effects is doubtful, particularly because other evidence suggests that ROS mediate the positive effects of exercise in other settings. Again, more elaboration is needed there.

Response: We appreciate your comments and have expanded the exercise section to include human and pig studies, and the underlying molecular mechanisms on ROS, eNOS function and cardiac transcription factor expression important to fetal heart development (lines 528-570).

v. some typos: l. 151 "rising" should be read; Table 1: some characters misplaced in the columns

Response: Corrected. Thank you.

Reviewer 2 Report

Reviewer response to Engineer et al, “Say NO to ROS…”, mdpi antioxidants

The review is well written and its value is merited.

The references used are both primary literature (majority) and earlier published reviews related to topics touched upon, but not directly preceedents to the present review. The problem addressed is highly important and the review deserves to be published, however, after major revision.

I wonder why no references are included on the major contributions of Prof Vanhoutte on NO and uncoupled eNOS?

I find that some amendments may increase the reviews impact and readability for the future audience.

First, I would re-structure the review a bit to condense and make the red thread even more clear. At present, connections between sections in the review are missing.  Many building blocks are included, but the house is not build.

Examples:
The section 2.2 is somewhat a repetition of the introduction, the two could be merged, or the introduction could be shortened (e.g. To highlight the numbers of incidences, perspectives and consequences of maternal diabetes and congenital heart defects for society) and thereafter include the section on risk factors.
I find it odd to see a section on how to generate a rat model of diabetes in the introduction (lines 52-58). Pops somewhat out of the blue. On the other hand – inclusion of the section is merited, but might be better placed in connection with a section highlighting evidence gathered from animal experiments.

This brings me to the next confusing issue:
it would be great that the authors could highlight and make explicit statements every time they refer to animal experiments, e.g. With statements on embryonic day XX… what species is referred to? A separate section on evidence gathered from animal experiments could help here, next to a section on evidence / studies on humans (which I agree very much with the authors are limited but highly needed).

Please highlight what evidence on e.g. NOS3 function is gathered from eNOS’ role in the vasculature, and then extrapolated to its function in heart tissue, and what direct evidence is to be found in literature regarding NOS3 and heart development (section 4.2)?

Section 3.1. Could be part of intro.
Use of “offspring” and “children” should be critically chosen. I would prefer “children” whenever authors are referring to findings on humans. Would increase ease of reading and understanding (cf above)

Already from figure 1 it is evident that eNOS, ROS and NO are interconnected and affect eachother, possibly readers would benefit from a merged figure? This of course requires merging of text sections as well (see above re thin read line) – Read flow could e.g. be 4.1, 4.4, 4.2, 4.3, 5.1??

Figure 3: I miss directy readable information from figure 3 that the figure regards CHD and the metabolic pathways highlighted are from the fetal heart… it might furthermore be an advantage to extend this figure to also highlight effects of maternal DM on transcription factors and thereby also on alterations of transcriptional activity (which I guess is the true reason for the develomental defects)

Authors may in their section on ROS want to highlight that H2O2 not always is the bad guy, at least not in host response. Furthermore, H2O2 plays a major role in e.g. Vasodilatation in diseased humans (see Gutterman, Shimokawa, De Mey works in patient arteries)

Any thoughts on the contributions of advanced glycations end products (AGEs) and receptor of AGEs (RAGE) may be a valuable contribution. I believe also the carbonyl stress has major implications on heart development, could be added to figure 3….

Section 6.1, may be beneficial to somehow merge with section 4.4. And with last lines of introduction in a new section (cf above)

Table 1 is a great addition and overwiew!

Section 7: beginning is (again) somehow a recap of the introduction…

Minor issues:
several abbreviations are not defined upon first use.
Table 1: readers may benefit from landscape format, especially regarding column 4 formatting.
Figure 1, right side (uncoupled eNOS), readers would benefit from larger (highlighting) ONOO and O2- red letters - sizes like NO in blue is large in the left side of the figure.
Authors may want to stream line the design of all three figures (letters, use of boxes and squares, arrows, colors, symbols (e.g. For eNOS) etc.)
Fig3: define abbrev in fig legend
Refs in conclusion may be redundant
Line 472 – I would add % lowering to highlight the research results.

Author Response

We are very happy to hear that “the problem addressed is highly important and the review deserves to be published”. We thank you for your insightful and constructive comments to improve our manuscript.

I wonder why no references are included on the major contributions of Prof. Paul Vanhoutte on NO and uncoupled eNOS?

Response: This is an oversight on our part. A recent review article of Prof. Paul Vanhoutte published in Circ Res 2016 is now cited as reference 97 (lines 198 and 219).

The section 2.2 is somewhat a repetition of the introduction, the two could be merged, or the introduction could be shortened (e.g. To highlight the numbers of incidences, perspectives and consequences of maternal diabetes and congenital heart defects for society) and thereafter include the section on risk factors.

Response: Introduction is now shorten as per your suggestion. The statement on risk factors is now moved to section 2.2.

I find it odd to see a section on how to generate a rat model of diabetes in the introduction (lines 52-58). Pops somewhat out of the blue. On the other hand – inclusion of the section is merited, but might be better placed in connection with a section highlighting evidence gathered from animal experiments.

Response: We fully agree with your comments and have now merged this section with a preceding paragraph.

It would be great that the authors could highlight and make explicit statements every time they refer to animal experiments, e.g. With statements on embryonic day XX… what species is referred to? A separate section on evidence gathered from animal experiments could help here, next to a section on evidence / studies on humans (which I agree very much with the authors are limited but highly needed).

Response: In response to your comments, animal species and human studies are now explicitly stated.

Please highlight what evidence on e.g. NOS3 function is gathered from eNOS’ role in the vasculature, and then extrapolated to its function in heart tissue, and what direct evidence is to be found in literature regarding NOS3 and heart development (section 4.2)?

Response: The role of NO and eNOS in the vasculature is briefly highlighted in Section 4.2.  Direct evidence regarding eNOS and heart development is described in section 4.4. A more in depth review solely on the subject of eNOS on heart development has been previously published by our group (Ref. 102) and is now mentioned at the end of this section (lines 308-310).

Section 3.1. Could be part of intro.

Response: In order to keep the Introduction brief, we prefer to leave Section 3.1 as is. We trust this is acceptable to you.

Use of “offspring” and “children” should be critically chosen. I would prefer “children” whenever authors are referring to findings on humans. Would increase ease of reading and understanding (cf above)

Response: Your recommendation is well taken. The word “children” is now used to refer findings on humans.

Already from figure 1 it is evident that eNOS, ROS and NO are interconnected and affect each other, possibly readers would benefit from a merged figure? This of course requires merging of text sections as well (see above re thin read line) – Read flow could e.g. be 4.1, 4.4, 4.2, 4.3, 5.1??

Response: We appreciate your comments. While merging the figures would save space, a merged figure that contains all components of the individual figures would be complicated and not easy to follow for readers. We therefore would prefer not to merge them, and hope this is acceptable to you.

Figure 3: I miss direct readable information from figure 3 that the figure regards CHD and the metabolic pathways highlighted are from the fetal heart… it might furthermore be an advantage to extend this figure to also highlight effects of maternal DM on transcription factors and thereby also on alterations of transcriptional activity (which I guess is the true reason for the developmental defects)

Response: Figure 3 summarizes possible mechanisms of hyperglycemia induced cellular oxidative damage in the fetal heart. Gene expression (due to transcription factor activation), cell proliferation, cell survival and EMT, which are critical to heart development are now explained in the figure legend.

Authors may in their section on ROS want to highlight that H2O2 not always is the bad guy, at least not in host response. Furthermore, H2O2 plays a major role in e.g. Vasodilatation in diseased humans (see Gutterman, Shimokawa, De Mey works in patient arteries)

Response: In response to your comments, the role of H2O2 as EDHF in regulating vascular function is now highlighted in section 5.1 and Shimokawa’s recent review on this topic is cited (see Ref. 145, Free Radic Biol Med 2017).

Any thoughts on the contributions of advanced glycations end products (AGEs) and receptor of AGEs (RAGE) may be a valuable contribution. I believe also the carbonyl stress has major implications on heart development, could be added to figure 3….

Response: AGEs, RAGE and carbonyl stress as a potential mechanism for CHDs are now included in sections 5.2 and 6.3, and Figure 3.

Section 6.1, may be beneficial to somehow merge with section 4.4. And with last lines of introduction in a new section (cf above)

Response: Section 6.1 describes animal models of CHDs induced by hyperglycemia and pregestational diabetes. Section 4.4 describes the role of eNOS in heart development without diabetes. They are on different topics, and we would prefer to leave them as they are. The last lines of Introduction are now merged with a preceding paragraph (see our Response above).

Table 1 is a great addition and overview!

Response: Thank you!

Section 7: beginning is (again) somehow a recap of the introduction…

Response: That sentence is now deleted.

Minor issues:

Several abbreviations are not defined upon first use.

Response: They are now defined upon first use.

Table 1: readers may benefit from landscape format, especially regarding column 4 formatting.

Response: Done.

Figure 1, right side (uncoupled eNOS), readers would benefit from larger (highlighting) ONOO and O2- red letters - sizes like NO in blue is large in the left side of the figure.

Response: Done.

Authors may want to stream line the design of all three figures (letters, use of boxes and squares, arrows, colors, symbols (e.g. For eNOS) etc.)

Response: Done.

Fig3: define abbrev in fig legend

Response: Done.

Refs in conclusion may be redundant

Response: They are now removed.

Line 472 – I would add % lowering to highlight the research results.

Response: Added on line 523. Thank you.

Round 2

Reviewer 1 Report

No further comment

Reviewer 2 Report

The authors did a great job revising their review. I congratulate them on their extensive and excellent work. I detected only at a few circumstances that "nitric oxide" was used in stead of the abbreviation NO, and some singularis/pluralis mistakes in the English writing, but honestly, these are so minor errors that I believe they will be caught during proof editing of the final manuscript, and I will not request a minor revision. I look forward to see the work online!